# Development and Characterization of Bio-Based Formaldehyde Free Sucrose-Based Adhesive for Fabrication of Plywood

**DOI:** 10.3390/polym16050640

**Published:** 2024-02-27

**Authors:** Longjiang Liu, Yongbo Jia, Lulu Zheng, Rui Luo, Hisham Essawy, Heming Huang, Yaming Wang, Shuduan Deng, Jun Zhang

**Affiliations:** 1Faculty of Chemical Engineering, Kunming University of Science and Technology, Kunming 650093, China; liulongjiang1206@126.com (L.L.); wym@kmust.edu.cn (Y.W.); 2School of Chemical Engineering, Yunnan Vocational College of National-Defense Technology, Yunnan Open University, Kunming 650223, China; 3Yunnan Provincial Key Laboratory of Wood Adhesives and Glued Products, Southwest Forestry University, Kunming 650224, China; luluzheng12345@163.com (L.Z.); 17869847191@163.com (R.L.); 4Department of Polymers and Pigments, National Research Centre, Dokki, Cairo 12622, Egypt; hishamessawy@yahoo.com; 5Kunming Xinfeilin Wood-Based Panel Group Co., Ltd., Kunming 650106, China; hhm12345@163.com

**Keywords:** sucrose, ammonium dihydrogen phosphate, esterification reaction, flame retardancy

## Abstract

In order to solve the problem of excessive consumption of petrochemical resources and the harm of free formaldehyde release to human health, biomass raw materials, such as sucrose (S) and ammonium dihydrogen phosphate (ADP) can be chemically condensed in a simple route under acidic conditions to produce a formaldehyde free wood adhesive (S-ADP), characterized by good storage stability and water resistance, and higher wet shear strength with respect to petroleum based phenolic resin adhesive. The dry and boiling shear strength of the plywood based on S-ADP adhesive are as high as 1.05 MPa and 1.19 MPa, respectively. Moreover, is Modulus of Elasticity (MOE) is as high as 4910 MPa. Interestingly, the plywood based on the developed S-ADP adhesive exhibited good flame retardancy. After burning for 90 s, its shape remains unchanged. Meanwhile, it can be concluded from thermomechanical analysis (TMA) and thermogravimetric analysis (TGA) that the S-ADP acquired excellent modulus of elasticity (MOE) and good thermal stability. It is thus thought promisingly that the use of S-ADP adhesive as a substitute for PF resin adhesive seems feasible in the near future.

## 1. Introduction

Adhesives play an important role in applications such as furniture, construction, electronic equipment, and the packaging industry [1]. As an important component of plywood, adhesives can directly affect the quality and performance of plywood [2]. Currently, formaldehyde based resins including urea formaldehyde (UF), phenol-formaldehyde (PF), and melamine-formaldehyde (MF)are widely used. Among them, PF is broadly employed for the fabrication of plywood due to advantages such as high bonding strength, good water resistance, high heat resistance, and chemical stability [3,4,5]. The consumption of petrochemical resources and the complication of the production process of PF resin, which produces harmful gases such as formaldehyde, imposed the necessity develop of green, non-toxic, and bio-based adhesives. Such systems attracted great attention from researchers, such as starch, tannins, glucose, lignin, and cellulose, which have been extensively with the purpose of replacing formaldehyde based adhesives [6,7,8,9,10,11,12]. 

Sucrose, as the main product of photosynthesis, is one of the most abundant organic compounds in the world, with advantages such as it being green, renewable, and having low cost, and wide sources [13,14,15,16,17,18]. Such features are the main reasons for its wide use in various chemical processing and food industries [19,20,21,22]. In addition, it is considered a highly oxygen-containing complex multifunctional molecule, containing active hydroxyl groups and two isomeric carbon atoms, which make it malleable for producing valuable materials [23]. The abundance of hydroxyl functional groups in the molecular structure of sucrose, which can serve as active reaction sites, renders it as an important raw material for preparing bio-based adhesives [24]. However, it contains more hydroxyl groups than those in sea oil, which requires modification to improve its water resistance. Song et al. [25] prepared a new type of water-resistant and environmentally friendly sucrose-based resin as wood adhesive starting from sucrose, hexanediamine, and ammonium persulfate. In this process, the sucrose was oxidized using ammonium persulfate for 45 min, and then reacted with hexanediamine to form a cross-linked network structure. The dry and wet strength values of the plywood prepared using this adhesive exceeded the Chinese national standard GB/T9846-2015 [25]. However, the prepared plywood did not show any resistance to boiling water with complete loss of strength, which is considered a critical issue considering the preparation process of this adhesive is relatively complicated whereas hexanediamine is a toxic material [26]. All these drawbacks are presenting serious limitations that hindered its further development. 

Therefore, it is very important to develop a new bio-based sucrose-based resin adhesive via a simple, cost-effective, and non-toxic process. In line, ADP is a low-cost and widely used inorganic compound, commonly used as a fire retardant for wood, fabrics, and paper, as well as a fertilizer, etc. [27,28,29,30]. ADP can decompose into phosphoric acid at high temperatures, phosphoric acid can undergo esterification reaction with sucrose or glucose degraded by sucrose to further condense leading to form a three-dimensional network system [24]. Furthermore, its hydroxyl group can interact with the hydroxymethyl groups in wood, which is likely to improve the shear strength of resulting plywood. Hence, the goal of this study is to employ sucrose and ADP for the preparation of a low-cost adhesive system with reasonable fire resistance in a facile route. The study will extend to the optimization and evaluation of the bonding performance, storage stability, water resistance, and fire retardancy. 

## 2. Experimental Section

### 2.1. Materials

Sucrose (S) with purity of 99% and ammonium dihydrogen phosphate (ADP) with purity of 99% were purchased from Shanghai McLean Biochemical Technology Co., Ltd. (Shanghai, China). Phenol (98%), lignin (L, 95% powder), cellulose (C, 95% powder), and formaldehyde (37% aqueous solution) were purchased from Kepler Biotechnology Co., Ltd. (Shanghai, China). Poplar (poplar L) veneers with a thickness of 2 mm and a moisture content of 8–10% was provided by Nanjing Panel Company (Nanjing, Jiangsu, China). 

### 2.2. The Preparation of S-ADP Adhesives and S-ADP Based Plywood

The detailed parameters and materials dosages required for the preparation of the adhesive are collected in Table 1. First, 20 g of S and 12 g distilled water were added to a three-neck flask under stirring for 1 min. After that, different amounts of ADP (referred to in Table 1) were added to sucrose (S) solution under stirring for 10 min at room temperature to obtain S-ADP1-3 adhesives, or 4 g of ADP was added into the S solution under stirring for 10 min at 80 °C to obtain the S-ADP4 adhesive. Meanwhile, 20 g of S and 12 g of distilled water were added to a three-neck flask under stirring for 10 min to obtain a neat S adhesive for comparison with the S-ADP adhesives. The detailed preparation process of the S-ADP4 adhesive is shown in Figure 1a. 

Meanwhile, PF adhesive was prepared according to reference [31]. The molar ratio of phenol to formaldehyde was set at 1:1.5. Firstly, sodium hydroxide, phenol, and water were mixed under stirring at 50 °C for 20 min. Afterward, a first part of the formaldehyde (accounting for 80% of the total formaldehyde) was added to the solution and left to react at 60 °C for 1 h, then the temperature was raised to 80 °C. Once the temperature is reached, the remaining formaldehyde was added (20% of total formaldehyde) and the reaction continued at 80 °C for 2 h until the viscosity attained a level of 300–700 mPa·s. 

Three-layer plywood samples were prepared according to the Chinese national standard (GB/T9846-2015), where three veneers with a size of 150 mm × 150 mm × 2 mm were dried in an oven at 100 °C for 24 h, followed by cooling at 20 °C in air with 65% humidity. The middle veneer layer (core board) was glued at a rate of 280 g/m^2^ on both sides then was tightly adhered to the other two layers of surface board, in which the adjacent two layers of wood board patterns were perpendicular to each other. The three-layer boards were assembled and performed under hot pressing at 200 °C for 6 min with a 5 mm thickness gauge. Finally, the prepared plywood was cut into six samples with dimensions of 100 mm × 25 mm × 6 mm for evaluation of mechanical performance. The process of plywood preparation is displayed in Figure 1b.

### 2.3. Characterizations 

The viscosity of the adhesive was measured at room temperature using a rotary viscometer (SNB-2, digital viscometer, Hengping Instrument Factory, Shanghai, China). The solid content test of the adhesive was determined by weighting a portion of the adhesive, which was placed in an oven (120 ± 1 °C) until it achieved complete curing. Then, it was taken out and cooled to room temperature before weighing again. The solid content was obtained from Equation (1):(1)$$M=\frac{W_{2}}{W_{1}}\times 100\%$$
where M is the solid content,%; W_1_ is the initial weight, g; W_2_ is the weight after drying, g.

The chemical structure of adhesives was studied using the fourier transform infrared spectroscopy (FTIR, Varian-1000, Palo Alto, CA, USA). The test sample was prepared by mixing 1 g of KBr with 0.01 g of sample powder, and the measurement was conducted over a wavenumber range of 400–4000 cm^−1^. Meanwhile, cellulose and/or lignin were mixed with the S-ADP4 adhesive in a mass ratio of 1:5 and cured in an oven at 200 °C, coded as S-ADP4-C and S-ADP4-L, respectively, and their chemical structures were also investigated.

The chemical structure of the adhesive was additionally analyzed using a triple quadrupole mass spectrometer (Waters Xevo, Milford, MA, USA) equipped with an electric spray ionization source (ESI-MS). The adhesive was dissolved in a concentration of 10 μL/mL of chloroform and the mixture was injected into an ESI source ion trap mass spectrometer (Brook Dalton, Buchareria, MA, USA) using a syringe at a flow rate of 5 μg/s. The spectrum was recorded over a scanning range of 1 Da to 2000 Da in a positive mode using ion energy of 0.3 eV.

The surface elements of the cured adhesive were investigated by X-ray photoelectron spectroscopy (XPS) of the NEXSA spectrometer (Thermo Fisher Scientific, Waltham, MA, USA). Furthermore, the crystallinity of adhesives was evaluated using an X-ray diffractometer (XRD, 7000, Hitachi, Tokyo, Japan) equipped with a CuKα radiation source with a wavelength of 1.5406 Å, while a scanning speed of 6°/min was employed over a range of 1–90°. Origin software (Northampton, MA, USA Z673Q-9089-7222222) was used to obtain the crystallization peak area (crystalline + amorphous), and the crystallinity of the samples was calculated according to Equation (2):(2)$$Crystallinity=\frac{area\;of\;crystalline\;peaks}{area\;of\;all\;peaks}\times 100\%$$

The curing behavior of the S-ADP adhesive was studied under nitrogen atmosphere at 30–250 °C at a heating rate of 10 K/min using a differential scanning calorimeter (DSC, 204 F1, Netzsch, Germany). In addition, the thermal performance of the S-ADP adhesive was studied using a thermomechanical analyzer (TMA, 242, Netzsch, Germany). For that, 0.125 g of S-ADP adhesive was applied to two poplar veneers (10 mm × 50 mm × 2 mm) and the run was undertaken over a temperature range of 30 to 300 °C at a heating rate of 5 K/min.

Likewise, the thermal stability of S-ADP adhesives was conducted using a thermogravimetric analyzer (TG, 209 F3, Netzsch, Germany) under nitrogen atmosphere, with a heating rate of 10 K/min in the temperature range of 30–900 °C. The temperature of the heat resistance index (THRI) of the sample was calculated according to Equation (3) [32,33]:*THRI* = 0.49 × [*T*5 + 0.6 × (*T30* − *T5*)](3)
where, T5 is the temperature at which the sample exhibited 5% mass loss; T30 is the temperature at which the sample encountered 30% mass loss. 

A cutting machine (455 AL, Lingtai, Suzhou, China) was used to cut the plywood into dimensions of 100 mm × 25 mm × 5 mm (as shown in Figure 1), then the shear strength was assessed using a universal testing machine (4476, Instron, Boston, MA, USA). The dry and wet shear strength values were determined according to the Chinese national standards GB/T17657-1999 and CB/T9846.7-2004, with a tensile rate of 2 mm/min and a maximum load of 5 N. The plywood was treated first before the wet strength test by immersion in water at 63 ± 3 °C for 3 h or in boiling water (100 °C) for 3 h. The final wet shear strength is 0.9 times of the original shear strength. In order to calculate the mean and standard deviation, 5 duplicate samples were subjected to dry shear strength and wet shear strength tests.

The cured adhesives were used for combustion testing using a butane torch (ZT-09, Iwatani Co., Ltd., Zhuhai, China) at the same flame temperature. Outside, the poplar veneer was cut into 80 mm × 10 mm × 2 mm, and which was soaked in the S-ADP4 adhesive for 5 min then cured at 200 °C for 6 min before the flame retardancy test using a ZT-09 butane torch. Similarly, the plywood based on the S-ADP4 adhesive was cut into 100 mm × 25 mm × 6 mm, and the flame retardancy test was conducted.

## 3. Results and Discussion

The viscosity and solid content of various sucrose-based adhesives are shown in Figure 1a. The neat S adhesive acquired a viscosity of approximately 46.7 mPa·s. It is worth noting that when ADP is added to S, the viscosity and solid content of the resulting S-ADPs adhesive increased. Specifically, when the content of ADP increases, the solid content of the corresponding adhesives increases. Moreover, the viscosity of S-ADP4 (92.7 mPa·s) is lower than that of S-ADP1 (112.6 mPa·s), indicating that S dissolved well into ADP at 80 °C for 10 min. Meanwhile, due to the release of some ammonia gas from ADP during the reaction at 80 °C for 10 min, the solid content was lower than that of S-ADP1. Figure 1b shows the appearance and viscosity changes of S, S-ADP1, and S-ADP4 adhesives over 10 days, which reveals that the appearance of the S-ADP4 adhesive remained unchanged during storage and maintained its white transparent appearance with a little change in viscosity, slightly increasing from 92 mPa·s to 99 mPa·s, indicating good storage stability. However, the appearance of the S-ADP1 adhesive showed no significant change and remained a white turbid solution with an increase in viscosity from 112.6 mPa·s to 120.6 mPa·s. However, small amounts of bubbles appeared between the 9th and 10th days, revealing that the storage stability of S-ADP1 began to be influenced from the 9th day onwards. Between the 9th and 10th days, there was increase in the viscosity of the S adhesive from 46 mPa·s to 63 mPa·s, indicating that from that time and is critical concerning storage. 

In order to better understand the elements content and bond strength of S, S-ADP1, and S-ADP4, XPS analysis was undertaken, and the results are shown in Figure 2. Figure 2a shows the type and content of elements for S, S-ADP1, and S-ADP4, which illustrates that the contents of N and P increase with the addition of ADP. Figure 2b–i shows the binding energies corresponding to different groups in the case of S, S-ADP1, and S-ADP4. In the case of C1s, 284.8 ev corresponds to C-C, whereas 283.2 ev and 286.3 ev represent C=O and C-O, respectively [34]. By comparing the C1s of S with S-ADP1 and S-ADP4, we can see that the relevant intensity of C=O and C-O peaks decreases after the addition of ADP. Meanwhile, In the case of O1s, the XPS spectra for S-ADP1 and S-ADP4 showed the O-C at a binding energy of 531.3 ev, while for O-P at 533.4 ev, N-H at 401.9 ev, and N-C at 400.3 ev [35]. These results confirm that cross-linking has been established between G and ADP.

Figure 3a–c show the X-ray diffractograms of S, S-ADP1, and S-ADP4, respectively. The crystallization peaks of S and S-ADP extend between 5-60°, with a noticeable broadening following the reaction between S and ADP, indicating that the addition of ADP to sucrose altered the crystallinity significantly during esterification and condensation steps. To calculate the crystallinity of sucrose in different resin systems, linear fitting of sucrose crystallization peaks in the diffractograms of S, S-ADP1, and S-ADP4 resins was employed. The crystallinity of S was estimated to be around 90.01%. In contrast, the crystallinity of S-ADP1 after the addition of ADP was around 22.94%, indicating that the addition of ADP and the consequent chemical reactions can significantly reduce the crystallinity of S. At the same time, the crystallinity of the S-ADP4 adhesive prepared by the-precross-linking reaction of S and ADP at 80 °C is reached only 12.73%. This indicates that the chemical reaction between ADP and S has been proven, further reducing its crystallinity.

The TG-DTG curves of various sucrose-based adhesives are shown in Figure 4a,b, and the pyrolysis rates of all samples are mainly divided into two stages according to the DTG traces, with corresponding parameters shown in Table 2. The first stage, except for S, occurred before 100 °C for all sucrose-based adhesives, while the second stage onset was around 300 °C. The thermal degradation before 100 °C is mainly signifying volatile impurities from the cured adhesive. The peak temperature of stage II in the case of S was around 289.93 °C, whereas the peak temperature of stage II in the case of S-ADP4 was attained at 298.78 °C. Although the temperature of stage II in the case of S-ADP4 is higher than that of S, its weight loss in stage II is 10.27%, which is much lower than the weight loss encountered in stage II of S, which is approximated around 41.42%. More importantly, at 800 °C, the remaining mass of S is 17.83%, which is much lower than the remaining mass in the case of S-ADP4 (63.71%). The THRI of S is higher than that of S-ADP4. Meanwhile, the remaining mass of S-ADP1 at 800 °C is close to that of S-ADP4, as both use the same amount of raw materials, resulting in similar residual carbon content. Meanwhile, the TG-DTG trends and THRI values of S-ADPs are basically consistent, indicating that the combination of ADP and S can improve the thermal stability due to reaction. Figure 4c depicts the MOE of the various adhesives, in which that of S reached 1400.4 MPa. However, it is worth noting that the addition of ADP increased the MOE, which reached 1977.6 MPa in the case of S-ADP2. On the contrary, the MOE of S-ADP1 was only 1927.1 MPa, indicating that an increase in ADP content can improve the toughness of the adhesive. The MOE of S-ADP4 (4910.3 MPa) is significantly higher than that of S-ADP3, indicating that pre-reaction of S and ADP at 80 °C for 10 min can promote their condensation and further improve the MOE of the resulting adhesive. In addition, the MOE of S-ADP4 (4915 MPa) is much larger than that of the widely used PF adhesive system (3000 MPa) [31], indicating its promising application prospects. 

Figure 4d shows the DSC curves of different sucrose-based adhesives. An exothermic peak can be observed for all adhesives. As the proportion of ADP in S increases, the curing temperature also increases. This indicates that the buildup of a network structure as a result of reaction between S and ADP becomes more complicated, requiring higher cross-linking temperatures. The curing temperature of S-ADP4 is lower than that of S-ADP1, indicating that pre-reaction of S and ADP at 80 °C for 10 min can improve the cross-linking degree of the resin system, reducing the curing temperature from around 172 °C in the case of S-ADP1 to around 162 °C in the case of S-ADP4. 

Compared to Figure 5, it was found in Figure 5b that sucrose reacts with ADP to produce polymers with different molecular weights, as revealed by peaks corresponding to molecular weights of 438 Da, 540 Da, and 1129 Da. These molecular weights reflect various degrees of condensation reactions between S and ADP, as corroborated by the peak at 1129 Da, which is relevant to an oligomer incorporating 2 of sucrose monomer, 1 of glucose monomer, and 4 of phosphoric acids. Firstly, ADP decomposed to yield phosphoric acid and ammonia. Meanwhile, S also pyrolysised into glucose. Due to its high reactivity, phosphoric acid undergoes a condensation reaction with the hydroxyl groups of S and glucose (Figure 5c), gradually increasing the degree of cross-linking between the molecules of sucrose, generating polymers with good structural stability and strong bonding interactions [24].

Figure 5d shows the FTIR spectra of cured S, S-ADPs adhesives, and S-ADP 4-C in comparison with L. The introduction of cellulose and lignin is accomplished to simulate the chemical bonding between the S-ADP4 adhesive and wood. The absorption peak at 3030–3600 cm^−1^ belongs to the stretching vibration of hydroxyl groups in sucrose. However, this peak appears more broader in the case of the S-ADP adhesive. In the range of 3030–3600 cm^−1^, there is an obvious shift between their peaks and the comparable peaks of the S adhesive, indicating that the addition of ADP alters its chemical environment via reaction of the hydroxyl groups of sucrose with ADP. Meanwhile, the absorption peaks of S-ADP4-C and S-ADP4-L at 3030–3600 cm^−1^ showed different states and shifts compared to the absorption peaks of S-ADP4, indicating that the cellulose and lignin focused their interaction on the hydroxyl groups in the S-ADP4 system, leading to changes in the chemical environment. In addition, the peak at 1600 cm^−1^ is a characteristic stretching vibration peak of -CHO in glucose. The C=O absorption peak of S is wider than that of S-ADP, which can be attributed to the changes in chemical environment caused by the reaction between sucrose and ADP. It can be observed that the absorption peaks referring to C=O became attenuated in the case of S-ADP4-C and S-ADP4-L after the insertion of cellulose and lignin, which can be correlated with its chemical interaction with these substrates via different interaction modes.

From Figure 6, it can be implied that the plywood based on the S adhesive does not present any strength because the veneer did not form a three-layer artificial board after pressing. This is totally expected due to the lack of a network structure, which is responsible for bearing the load. On the contrary, the dry strength, 63 °C hot water strength, and boiling water strength of S-ADPs-based plywood all met the Chinese national standard GB/T9846.3-2004 (≥0.70 MPa). The hot and boiling water strength values of S-ADP2 related plywood (1.05 MPa 1.04 MPa) are higher than the corresponding values of S-ADP1 related plywood. More importantly, the hot water strength (1.19 MPa) and boiling water strength (1.19 MPa) of S-ADP4 based plywood are higher than those formulated from S-ADP1-3, indicating that pre-reaction of S and ADP at 80 °C for 10 min can improve the cross-linking degree, further enhancing the adhesive performance. Figure 6b presents the shear strength of plywood prepared with some commercially available adhesives. The shear strength of S-ADP4 plywood (1.19 MPa) is higher than that of ammonium persulfate modified sucrose oxide plywood (OSUH, 1.14 MPa) [25], PF (0.91 MPa) [31], sucrose dihydrogen phosphate dephenol cottonseed protein (SADP-DCP, wet water strength 0.98 MPa) [36], and tannin-furanic (TF, 0.55 MPa) [37] adhesives, This indicates that the condensation of ADP and S could significantly enhance the water resistance of S-ADP. The appearance of plywood based on the GS-ADP adhesive after testing of wet shear strength (boiling water) is shown in Figure 6c, in which a significant wood damage was observed in S-ADPs plywood, indicating that the adhesive has very strong adhesive properties.

As demonstrated in Figure 7, the plywood based on the S-ADP4 adhesive is flame retardant. The shape of the S-ADP4 plywood remained unchanged and undamaged after 90 s combustion, while the shape of the S-ADP1 plywood underwent slight changes during 90 s combustion but did not show significant damage. This proves that cross-linking from the reaction of S with ADP can improve the flame retardancy of the resulting adhesive. In addition, pre-reaction at 80 °C for 10 min also has an improving effect on the flame retardancy. Meanwhile, it can be seen that the wood chip, which was soaked in prior in S-ADP1 and S-ADP4 adhesives, still maintained their original shape and exhibited not-intensive fractures after 90 s of combustion. Importantly, the wood chip that did not exposed to prior soaking in the S-ADP adhesive have turned into ash after only 20 s of combustion. Such results confirm that the S-ADP adhesive has effectively assisted in providing the wood with good flame retardancy, which opens the way in the future for good application prospects of flame retardant wood.

## 4. Conclusions

A Simple, facile, bio-based sucrose-based adhesive, named S-ADP adhesive, can be prepared under broad conditions via a condensation reaction between sucrose and ADP. This allows for good control of the viscosity and solid content of the resulting adhesives. Interestingly, the developed adhesives exhibited good storage stability and excellent MOE. Moreover, the addition of ADP improves the thermal stability of sucrose. More importantly, the distinct chemical activity of the formed network structures of the adhesives provided good dry as well as wet (hot and boiling water) strength values, all of which succeeded to meeting the Chinese national standard (GB/T9846.3-2004 ≥ 0.70 MPa). The wet shear strength of the S-ADP adhesive is even higher than that of the PF adhesive. Additionally, one of the most attractive features is the gained high flame retardancy. The plywood prepared based on the S-ADP adhesive maintains its original shape after combustion, which opens up new avenues for the application of S-ADP adhesive.

## Data Availability

Data are contained within the article.

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
