# Peer review of "Development and Characterization of Bio-Based Formaldehyde Free Sucrose-Based Adhesive for Fabrication of Plywood"

_polymers, 2024, doi:10.3390/polym16050640_

Round 1
Reviewer 1 Report
Comments and Suggestions for Authors
Lines 58. ADP can undergo esterification reaction with sucrose to further condense leading to formation of a three-dimensional network system [24]. The results of [24] must be described more comprehensively and should be used while discussing structures of the S-ADP.
H3PO4 is a well-known substance increasing the charcoal yield in the processes of wood pyrolysis. This fact should be taken into account in the manuscript.
Line 75. in Table 1. First, 20g of S and 12g distilled water were added … But 6 ml of DL is given in Table 1.
There is the contradiction between Line 174 (the viscosity of S-ADP4 (196mPa. s)) and line 181 (S-ADP4 … with a viscosity, slightly increasing from 92 mPa·s to 99 mPa·s)…
Lines 262-264. These molecular weights reflect various degrees of condensation reactions between S and ADP, as corroborated by the peak at 1297 Da, which is relevant to a polymer incorporating 2 mol sucrose and 4 mol ADP.
First, a polymer incorporating 2 mol sucrose and 4 mol ADP is not a polymer, and moreover is not a compound!
Second, this “compound” must have even but not odd molecular weight.
The third, MM of the compound is of 1008, but not 1297.
The forth, peak 1297 lacks in Fig. 6.
Fig. 6, a, insertion, sucrose molecule, 956 and M+Na. molecular mass of sucrose is of 342, but not 956. What is Na?
Finally, there are no any arguments in the manuscript confirming the main structure of the “compound” in Fig.6.
Fig. 7 permits to compare the shear strength of plywood based on sucrose based-adhesives, and based on some reported adhesives. But, Fig 8 does not contain data which allow similar comparing retardancy of the suggested and reported plywood samples.
Author Response
1. Lines 58. ADP can undergo esterification reaction with sucrose to further condense leading to formation of a three-dimensional network system [24]. The results of [24] must be described more comprehensively and should be used while discussing structures of the S-ADP.
Reply to reviewer: Thank you for the reviewer's comments. Reference 24 does not provide a detailed description of the reactions between S and ADP. In our article, the detail reactions between S and ADP are shown in Figure 6c. We have also described them according to reference 24 in the introduction and discussion sections, marked it with green highlighting
2. H3PO4 is a well-known substance increasing the charcoal yield in the processes of wood pyrolysis. This fact should be taken into account in the manuscript.
Reply to reviewer:Thanks for the reviewer's comments. We also considered this issue before the experiment, but later we found that phosphoric acid and sucrose can react quickly at high temperatures. The prepared plywood did not have any quality issues and had high shear strength.
3. Line 75. in Table 1. First, 20g of S and 12g distilled water were added … But 6 ml of DL is given in Table 1.
Reply to reviewer:Table 1 has been changed and marked it with yellow highlighting
4.There is the contradiction between Line 174 (the viscosity of S-ADP4 (196mPa. s)) and line 181 (S-ADP4 … with a viscosity, slightly increasing from 92 mPa·s to 99 mPa·s)…
Reply to reviewer:Thank you for the careful observation of the reviewer. We made a serious mistake and created two images at the same time. We put the other image in the article. We have now made the corresponding changes to the image and revised the description accordingly and marked it with gray highlighting.
5. Lines 262-264. These molecular weights reflect various degrees of condensation reactions between S and ADP, as corroborated by the peak at 1297 Da, which is relevant to a polymer incorporating 2 mol sucrose and 4 mol ADP.First, a polymer incorporating 2 mol sucrose and 4 mol ADP is not a polymer, and moreover is not a compound! Second, this “compound” must have even but not odd molecular weight. The third, MM of the compound is of 1008, but not 1297.The forth, peak 1297 lacks in Fig. 6.
Reply to reviewer: Thank you again for the careful review by the reviewer. Indeed, we have used mass ratio instead of molar ratio. We have carefully revised according to the reviewer's question. Firstly, we have changed the polymer to oligomer. Secondly, using molar ratio is incorrect. We have made the necessary changes and marked it with blue highlighting. In addition, we carefully checked the data and found that its corresponding peak is 1129 Da (composed of 2 sucrose, 1 glucose, and 4 phosphate) instead of 1297 Da, which we have corrected (Because it adds a hydrogen, it is an odd number). And we have also put the structure of the corresponding oligomers in Figure 6b.
6. a, insertion, sucrose molecule, 956 and M+Na. molecular mass of sucrose is of 342, but not 956. What is Na?
Reply to reviewer: Thank you for the reviewer's advise. We made a calculation error and have now put the correct structure corresponding to 956 Da in Figure 6a. In addition, All the peak values are based on the molecular weight (MW) of the species + 23 Da due to the Na+ of the NaCl matrix enhancer used, or + 1 Da due to H+ protonation.
7.Finally, there are no any arguments in the manuscript confirming the main structure of the “compound” in Fig.6.
Reply to reviewer: The structure of the oligomer in Figure 6b (1129Da) can be derived from the peak data of ESI-MS, and the reaction formula in Figure 6c can be randomly derived based on the date and the possibility of S and ADP reactions.
8. Fig. 7 permits to compare the shear strength of plywood based on sucrose based-adhesives, and based on some reported adhesives. But, Fig 8 does not contain data which allow similar comparing retardancy of the suggested and reported plywood samples.
Reply to reviewer: Thanks for the reviewer's comments. After reviewing most of the literature, we have not found any flame retardant tests for bio-based or petroleum-based adhesives. However, compared to wood, the plywood based on S-ADP adhesive we prepared still maintains its original state after combustion, indicating that it has flame retardancy.

Reviewer 2 Report
Comments and Suggestions for Authors
This article is devoted to the production and study of a new wood glue based on sucrose and ammonium dihydrogen phosphate. The authors claim that this development can be a good alternative to formaldehyde adhesives for wood. The relevance of the study does not raise any questions, since the development of environmentally friendly methods and approaches to new technologies is an important task. The article is well written and the main ideas are clear. There are some points that need to be improved:
1. The Abstract needs to be expanded by adding some data from the article. Now this looks very general.
2. Most of the experimental data should be compared with data from literature sources.
3. It is advisable to have a comparison experiment, where the classical approach with formaldehyde is studied. As a last resort, a more detailed comparison with data from literature sources was carried out.
4. The article contains spelling and stylistic errors. It would be advisable to fix this.
5. The conclusions look very modest. They need to be expanded.
Author Response
- The Abstract needs to be expanded by adding some data from the article. Now this looks very general.
Reply to reviewer:Thanks for the reviewer's comments. We have added some important data to the abstract,and marked it with yellow highlighting.
- Most of the experimental data should be compared with data from literature sources.
Reply to reviewer: Thanks for the reviewer's comments. We have compared the most important bonding performance data with other literature, and we have not found any tests related to flame retardancy that are mentioned in other literature.
- It is advisable to have a comparison experiment, where the classical approach with formaldehyde is studied. As a last resort, a more detailed comparison with data from literature sources was carried out.
Reply to reviewer:Thanks for the reviewer's suggestion. The adhesive we prepared is a formaldehyde free adhesive, so we did not compare the relevant formaldehyde release issues.In the article we have a comparison of the shear strength of formaldehyde-based adhesive such as PF adhesive as well as other biomass adhesives.
- The article contains spelling and stylistic errors. It would be advisable to fix this.
Reply to reviewer: Thanks for the reviewer's comments. We have made every effort to correct the grammar and spelling errors in the article.
- The conclusions look very modest. They need to be expanded.
Reply to reviewer:Thanks for the reviewer's comments. We have made some modifications to the conclusion, and marked it with yellow highlighting.

Reviewer 3 Report
Comments and Suggestions for Authors
The present paper by Liu and co-workers investigates a new plywood adhesive (S-ADP), formed by sucrose and dihydrogen phosphate. The preparation method is carefully described, and this adhesive is characterized by various methods such as TG, XRD, FT-IR. Its adhension strength is shown to be promising based on shear test results. In general, it is an interesting paper that provides deep insights into the understanding of this new adhesive, and shows promising applications. I recommend this paper to be published in Polymers after proper responses to the question below.
In Figure 6, the FT-IR spectra show heavy baseline; interpretation may be impacted by the quality of these spectra. Also, looks like some of them are represented in transmittance whereas the others are represented in absorbance; it's better to keep consistent.
Overall English is good for publication; minor typo correction and grammartically editing is needed.
Author Response
- In Figure 6, the FT-IR spectra show heavy baseline; interpretation may be impacted by the quality of these spectra. Also, looks like some of them are represented in transmittance whereas the others are represented in absorbance; it's better to keep consistent.
Reply to reviewer: Thanks for the reviewer's comments. All FTIR curves are based on a single pattern of absorbance, and although some curves may seem a bit strange, they do not affect to obtain the required characteristic peaks. Meanwhile, we adjusted these curves and put the modified image in Figure 6.
- Overall English is good for publication; minor typo correction and grammartically editing is needed.
Reply to reviewer: Thanks for the reviewer's comments. We have made every effort to correct the grammar and spelling errors in the article.

Round 2
Reviewer 1 Report
Comments and Suggestions for Authors
Dear Authors,
please, see two my remarks below.
7. Finally, there are no any arguments in the manuscript confirming the main structure of the “compound” in Fig.6.
Reply to reviewer: The structure of the oligomer in Figure 6b (1129Da) can be derived from the peak data of ESI-MS, and the reaction formula in Figure 6c can be randomly derived based on the date and the possibility of S and ADP reactions.
These are not convincing arguments. Main peaks of spectra 6a and 6b (274, 338, and 675) coincide, and, hence, main part of spectrum 6b does not connected with H3PO4 addition.
And about Table 2 and Fig. 5a. Why Authors did not try to explain coincidence of lost mass at 800 oC (63%) for S-ADP1 - S-ADP4 samples?
Author Response
1.These are not convincing arguments. Main peaks of spectra 6a and 6b (274, 338, and 675) coincide, and, hence, main part of spectrum 6b does not connected with H3PO4 addition.
Reply to reviewer: Thank you for the reviewer's comments, but the peaks corresponding to S-ADP oligomers are at 438 Da, 540 Da, and 1129 Da,which shown in Fig. 6b, not the 274 Da, 338 Da, and 675 Da mentioned by the reviewer. The 274 Da, 338 Da, and 675 Da peaks mentioned by the reviewer are the existing structures of sucrose, which also appear in Figure 6b due to the presence of unreacted sucrose and related structures.
- And about Table 2 and Fig. 5a. Why Authors did not try to explain coincidence of lost mass at 800 oC (63%) for S-ADP1 - S-ADP4 samples?
Reply to reviewer: Thank you for the reviewer's comments. We have provided a reasonable explanation in the article and marked it with yellow highlighting.
Reviewer 2 Report
Comments and Suggestions for Authors
Accepted
Author Response
Thanks to the reviewer for accepting our article